# Rsph4a is essential for the triplet radial spoke head assembly of the mouse motile cilia

**Hiroshi Yoke**[1☯¤], **Hironori Ueno**[2☯], **Akihiro Narita**[3☯], **Takafumi Sakai**[1☯], **Kahoru Horiuchi**[1], **Chikako Shingyoji**[1], **Hiroshi Hamada**[4], **Kyosuke Shinohara**[1]*

1 Department of Biotechnology & Life Science, Tokyo University of Agriculture & Technology, Koganei, Tokyo, Japan, 2 Molecular Function & Life Sciences, Aichi University of Education, Kariya, Aichi, Japan, 3 Structural Biology Research Center, Graduate School of Science, Nagoya University, Nagoya, Aichi, Japan, 4 Center for Biosystems Dynamics Research, RIKEN, Kobe, Japan

☯ These authors contributed equally to this work.
¤ Current address: National Institute for Basic Biology, Okazaki, Aichi, Japan
* k_shino@cc.tuat.ac.jp

**Data Availability Statement:** All relevant data are within the manuscript and its Supporting Information files.

## Abstract

Motile cilia/flagella are essential for swimming and generating extracellular fluid flow in eukaryotes. Motile cilia harbor a 9+2 arrangement consisting of nine doublet microtubules with dynein arms at the periphery and a pair of singlet microtubules at the center (central pair). In the central system, the radial spoke has a T-shaped architecture and regulates the motility and motion pattern of cilia. Recent cryoelectron tomography data reveal three types of radial spokes (RS1, RS2, and RS3) in the 96 nm axoneme repeat unit; however, the molecular composition of the third radial spoke, RS3 is unknown. In human pathology, it is well known mutation of the radial spoke head-related genes causes primary ciliary dyskinesia (PCD) including respiratory defect and infertility. Here, we describe the role of the primary ciliary dyskinesia protein Rsph4a in the mouse motile cilia. Cryoelectron tomography reveals that the mouse trachea cilia harbor three types of radial spoke as with the other vertebrates and that all triplet spoke heads are lacking in the trachea cilia of Rsph4a-deficient mice. Furthermore, observation of ciliary movement and immunofluorescence analysis indicates that Rsph4a contributes to the generation of the planar beating of motile cilia by building the distal architecture of radial spokes in the trachea, the ependymal tissues, and the oviduct. Although detailed mechanism of RSs assembly remains unknown, our results suggest Rsph4a is a generic component of radial spoke heads, and could explain the severe phenotype of human PCD patients with *RSPH4A* mutation.

## Author summary

Motile cilia are nanodevices driving extracellular fluid flow and are involved in human primary ciliary dyskinesia (PCD) including respiratory diseases, infertility, and laterality defect. Radial spoke (RS) is a T-shaped architecture inside of the axoneme of motile cilia and it regulates the motility and motion pattern of cilia. RS consists of the spoke head and the stalk, and the three-types of RS (RS1, RS2, RS3) exist in the axoneme of motile cilia.

**Funding:** This work was supported by Core Research for Evolutional Science and Technology (CREST), Japan Science and Technology Corporation (JST) (project no. JPMJCR13W5 to H. H.), and by the Asahi Glass Foundation (project no. 1 to K.S.). The funders had no role in study design, data collection and analysis, decision to publish, or preparation of the manuscript.

**Competing interests:** The authors have declared that no competing interests exist.

To date, it is well known mutation of the spoke head-related genes causes PCD. Among the spoke head-related genes, mutation of *RSPH4A* leads to most severe phenotype on the cilia ultrastructure in the PCD patients, but it remains unknown what determines the severe phenotype. Here, we show the role of the primary ciliary dyskinesia causal gene *Rsph4a* in the mouse motile cilia. We have found that Rsph4a plays a central role in all the three-types of RS assembly and is a generic component of spoke heads in the trachea, the ependymal tissues, and the oviduct in the mouse. Our results could explain the severe phenotype of human PCD patients with *RSPH4A* mutation.

## Introduction

Primary ciliary dyskinesia (PCD) is a recessive genetic disease caused by defects in motile cilia function. To date, numerous causal genes have been identified in PCD patients [1]. Typical PCD causal genes are involved in the assembly of the axonemal dynein complex of human motile cilia [2–10]. In mice and humans, multiple motile cilia exist in the trachea, brain/ependymal, oviduct, inner ear, nasal, and testis. The mouse multiple motile cilia have a 9+2 type geometry that contains nine peripheral doublet microtubules with dynein arms, single microtubules at the center of the axoneme (central pair; CP), and radial spokes (RSs). CPs and RSs cooperatively control dynein activity via a mechanochemical interaction [11–13]. In addition to the axonemal dynein-related genes, deficiency of the RSs-related proteins also causes the PCD phenotype in humans [14–17]. Circular motion rather than planar beating of respiratory cilia is observed in human PCD patients who harbor RSPH1, RSPH4A, and RSPH9 mutations [14, 18, 19]. Furthermore, Frommer et al. found that RSPH4A rather than RSPH1 and RSPH9 plays a central role in radial spoke head assembly by immunofluorescence analyses of respiratory cilia in PCD patients [20]. In the patients, various ultrastructural defects of respiratory cilia was observed including translocation of outer doublet into the center, absence of central pair, single microtubule in the center, extra central microtubule, extra outer microtubule [14]. The proportion of respiratory cilia with normal axonemal structure is 50% in human *RSPH4A* patients [18], whereas it is 80% in RSPH1 patients [16], suggesting that the phenotype of the RSPH4A mutation is more severe than the RSPH1 mutation. Another RSs-related protein, RSPH3 is critical for the assembly of radial spoke in the human respiratory cilia and its mutation causes PCD [21]. Rsph6a is essential for the assembly of mouse sperm flagella and fertility [22].

In terms of structure, RSs are beneficial architecture. Most eukaryotic species, except for *Chlamydomonas* and *S. bulatta*, have the three types of RSs (RS1, RS2, RS3) within the 96 nm axoneme repeat unit [23, 24]. The RSs maintain evolutionarily conserved T-shaped morphology but have distinct detailed ultrastructures. In *Chlamydomonas*, RS1 and RS2 show similar ultrastructures. The spoke heads look like a parallelogram plate in a two-fold rotational symmetry. RS3 is missing, but the base and part of the stalk (called RS3-S) are retained [25]. Metazoa, sea urchin sperm, zebrafish sperm, and human respiratory cilia show triplet RSs revealed by cryoelectron tomography (cryo-ET) [17, 26, 27]. Interestingly, RS3 is unaffected in human PCD patients with RSPH1 mutations, suggesting that the molecular composition is distinct among the three types of RSs [17]. Thus, the molecular basis of RS3 remains unknown [17, 28]. In this work, we examined the structure of RSs in mouse motile cilia by cryo-ET and immunofluorescence. Using wild-type (WT) mice and *Rsph4a* KO mice, we found that Rsph4a is essential for the assembly of the RS heads of the three types of RSs, and deficiency of

Rsph4a leads to typical PCD phenotypes due to the abnormal motion pattern of the mouse motile cilia in the trachea, brain, and oviduct.

## Results

### Rsph4a regulates the motion pattern of mouse motile cilia

In a previous study, Shinohara et al. reported that the ciliary motion pattern of trachea cilia was disorganized in *Rsph4a* KO mice. The cilia showed clockwise rotation motion rather than planar beating [29]. The *Rsph4a* KO mice show hydrocephalus which is a typical phenotype of PCD (Fig 1A). To study the comprehensive role of Rsph4a in mice, we examined the motion of the ependymal cilia in the subventricular zone and the oviduct cilia in addition to the observation of the trachea cilia. In the trachea, we again observed a change in the motion pattern in *Rsph4a* KO mice. The trachea cilia show clockwise rotation, whereas they show planar beating in the WT mice (Fig 1B, S1 Video, S2 Video), and we have confirmed the reproducibility of previous observations [29]. The phenotype is different from that in the trachea cilia in both in the ependymal cilia and the oviduct cilia. In the WT mice, all the ependymal cilia and the oviduct cilia show planar beating (Fig 1C and 1D; N = 80 cells, S3 Video, S5 Video). In Rsph4a KO mice, all the ependymal cilia show irregular motion, including rotation and wavy motion (Fig 1C N = 60 cells, S4 Video). The oviduct cilia show the two types of motion patterns, including anti-clockwise rotation (27%, N = 52 cells) and beating with small amplitude (73%, N = 52 cells) in Rsph4a KO mice (Fig 1D, S6 Video, S7 Video). Our observations suggest that Rsph4a regulates the motion pattern of the mouse motile cilia, although the phenotype is different among the cell types.

### Cryoelectron tomography revealed the ultrastructure of the mouse motile cilia

To address the mechanism of regulation of ciliary motion pattern and the role of Rsph4a protein, we next examined the ultrastructure of mouse motile cilia by cryo-ET. We analyzed the ultrastructure of the mouse trachea cilia because it is possible to isolate and collect trachea cilia for cryo-ET [30]. We dissected the mouse trachea and delicately rubbed it onto the wall of the tube to isolate cilia. Then, trachea cilia are frozen in liquid ethane and observed by a cryoelectron microscope (cryo-EM) [30] (S1 Fig & Materials and methods). By subtomogram averaging, the ultrastructure of the 96 nm repeating unit of axoneme was visualized (Fig 2A–2C). In the 96 nm axoneme unit of the mouse trachea cilia, four outer dynein arms with two heads (pink), seven types of inner dynein (purple) arms, and a dynein regulatory complex (N-DRC; yellow) were observed (Fig 2A & 2C). Resolution of the averaged structure of the 96 nm axonemal repeat is 4.5 nm (Fourier shell coordination = 0.5, Fig 2D). These results suggest that the structure and arrangement of dynein arms of mouse trachea cilia are quite similar to those in human respiratory cilia (Fig 2E, [17]) and in zebrafish sperm (Fig 2F, [27]). On the RSs, however, there is a distinct feature compared with RSs in the other vertebrates. In RS3, the spoke head is more compact than that in human respiratory cilia, and the physical contact between RS2 and RS3 was not observed in the mouse trachea cilia. Alternatively, an axial protrusion was observed at the proximal side of the radial spoke head of RS3, and this architecture was physically close to the neck/arch of RS2 (Fig 2A and 2B). The protrusion was also observed at the base of radial spokes in sea urchin sperm [26]. In terms of the standing angle to the doublet microtubule (Fig 3A–3D), RS3 has a unique feature: the angle of the spoke head-stalk axis is different between RS1/RS2 and RS3 because the stalk of RS3 shows bending at the base (Fig

WT    *Rsph4a* KO

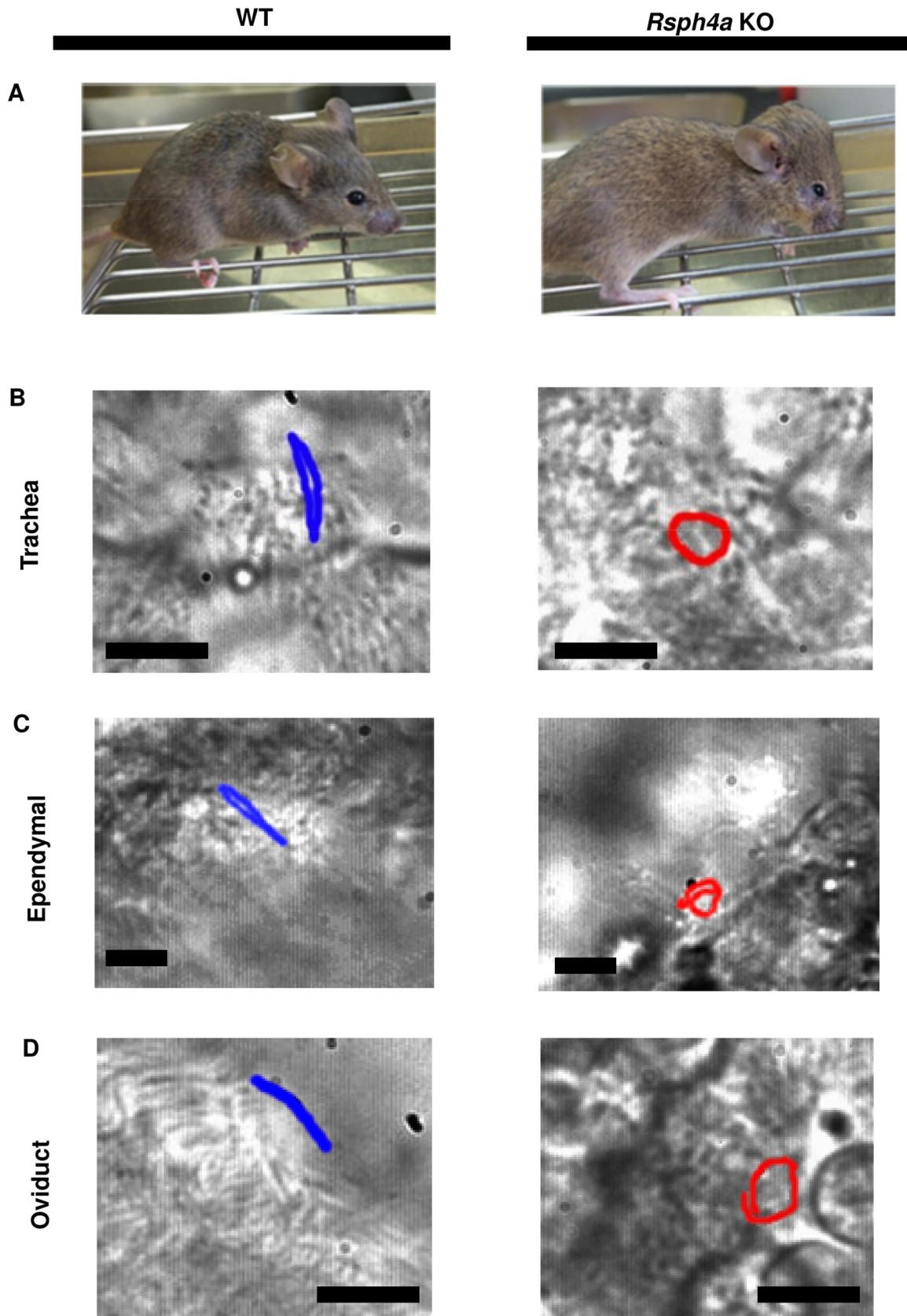

**Fig 1. Phenotype of the Rsph4a KO mouse. A**, Overview of the wild-type mice (WT; left) and the Rsph4a KO mice (right). The Rsph4a KO mice show hydrocephalus. **B** Ciliary tip motion in the trachea. The cilia show planar beating and clockwise rotation in the WT mice and the Rsph4a KO mice, respectively. **C** Motion of the cilia in the ependymal cell (brain). The cilia show planar beating and clockwise rotation in the WT mice and the Rsph4a KO mice, respectively. **D** Motion of the cilia in the oviduct. The cilia show planar beating and anticlockwise rotation in the WT mice and the Rsph4a KO mice, respectively. All the size bars indicate 5 μm.

3D). The cryo-ET data of the WT trachea cilia suggest that RS2 and RS3 share similar morphological features, but the base and stalk architectures differ from each other.

## Rsph4a is essential for triplet radial spoke head assembly in the mouse motile cilia

We next examined the effect of Rsph4a deficiency on the ultrastructure of the mouse trachea cilia. In Rsph4a KO mice, all three types of spoke heads are missing, suggesting that Rsph4a plays a critical role in triplet spoke head assembly (Fig 4A and 4B, S2 Fig). Unexpectedly, furthermore, the spoke head and the neck/arch were missing in each RS in *Rsph4a* KO mouse. (Fig 4C–4E). To validate these findings, we examined the subcellular localization of radial spoke head proteins by immunostaining (Fig 5, Fig 6, Fig 7). Rsph4a localized in the trachea cilia of the WT mice but was lost in Rsph4a KO mice (Fig 5A–5F). Ciliary localizations of Rsph4a were missing in Rsph4a KO mice both in the ependymal cells (brain) and the oviduct cells (Fig 6A–6F, Fig 7A–7F). We next examined the localization of the two kinds of spoke head homolog proteins, Rsph9 and Rsph1. In the WT mice, Rsph9 and Rsph1 were localized in the trachea cilia, the ependymal cilia, and the oviduct cilia (Fig 5G–5I & 5M–5O, Fig 6G–6I & 6M–6O, Fig 7G–7I & 7M–7O). In Rsph4a KO mice, however, the ciliary localization of Rsph9 was dramatically reduced in the tissues (Fig 5J–5L, Fig 6J–6L, Fig 7J–7L).While ciliary localization of Rsph1 is reduced in the oviduct (Fig 7P–7R), it retains in the trachea and the ependymal cells of *Rsph4a* KO mice (Fig 5P–5R, Fig 6P–6R). To examine the level of protein, we carried out western blotting of these proteins (S3 Fig). In the trachea, we observe significant difference of protein level of Rsph1 between the wildtype and *Rsph4a* KO mice. The immunofluorescence data and the western blotting data suggest that Rsph4a is essential for the assembly of the spoke head complex in the mouse motile cilia. We finally examined the localization of Rsph23, a homolog of *Chlamydomonas* neck/arch protein Rsp23 [20, 25, 31–34]. A very recent work reports that Mutation of Rsph23/NME5 leads to PCD phenotype in Alaskan Malamutes [34]. In the WT mice, ciliary localization of Rsph23 was observed in the trachea, ependymal, and oviduct cells (Fig 5S–5U, Fig 6S–6U, Fig 7S–7U). Conversely, Rsph23 was not localized in the axoneme of the motile cilia in the ependymal tissue and the oviduct tissues in *Rsph4a* KO mice (Fig 6V–6X, Fig 7V–7X). In the trachea, weak staining of Rsph23 retained in the ciliated cells of *Rsph4a* KO mice (Fig 5V–5X). To validate the difference of level of protein, we carried out western blotting using the trachea tissues and we observed significant difference of protein level of Rsph23 between the wildtype and *Rsph4a* KO mice (S3 Fig). The western blotting data indicate that Rsph23 was reduced in the trachea cells of Rsph4a KO mice (S3 Fig). Our immunofluorescence data as well as the cryo-ET data suggest that the spoke head and the neck/arch are disrupted in the absence of Rsph4a in the mouse motile cilia.

## Discussion

Previous works have revealed that the morphology of RS3 is different from that of RS1 and RS2 [17, 25]. Additionally, in the mouse trachea cilia, the morphology of the stalk of RS3 is unique compared with that of RS1 and RS2 (Fig 3). On the other hand, all triplet spoke heads utilize Rsph4a as a common building block (Fig 4). The triplet spoke heads are absent in the

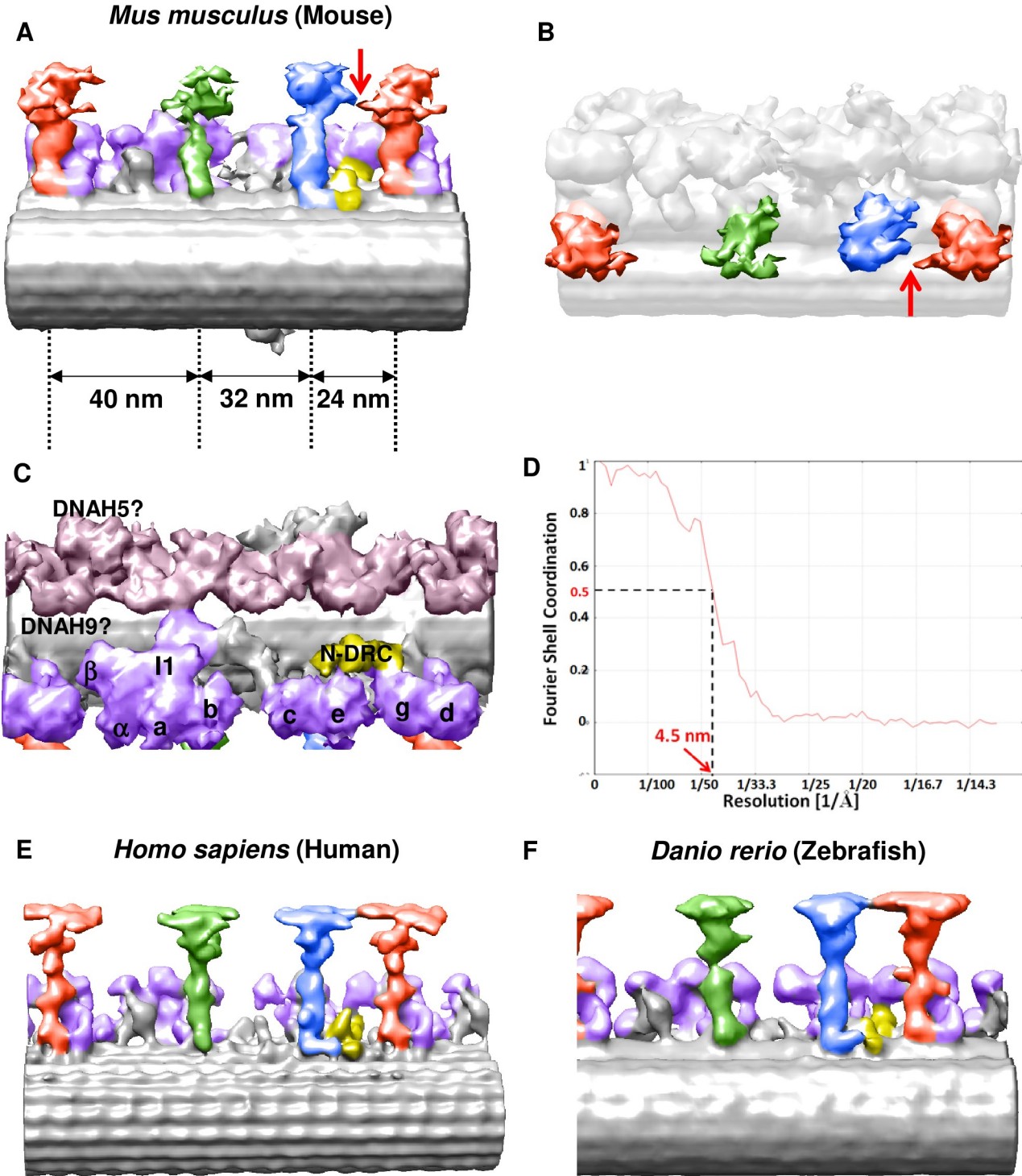

**Fig 2. Cryoelectron tomography of radial spokes (RSs) of the mouse trachea cilia. A**, Axial view of the trachea cilia. The mouse trachea cilia have triplet RSs, including RS1 (green), RS2 (blue), and RS3 (red). The interspoke distances are 40 nm (RS3-RS1), 32 nm (RS1-RS2), and 24 nm (RS2-RS3). **B** Overview of the triplet RSs. The spoke heads have a skate blade-like morphology in RS2 and RS3. An axial protrusion of RS3 closes to the neck/arch of RS2 (shown in an arrowhead). **C** Axonemal dyneins. The outer arm (pink) has two types of heads. Seven kinds of inner dynein (purple) and dynein regulatory complex (N-DRC; yellow) exist in the 96 nm repeat unit. **D** Resolution of the averaged structure of the 96 nm axonemal repeat. (Fourier shell coordination = 0.5). **E** Cryo-EM structure of the axonemal repeat of the human respiratory cilia (EMD-5950) in the paper by Lin et al. (2014). **F** Cryo-EM structure of the axonemal repeat of the zebrafish sperm (EMD-6954) in the paper by Yamaguchi et al. (2018).

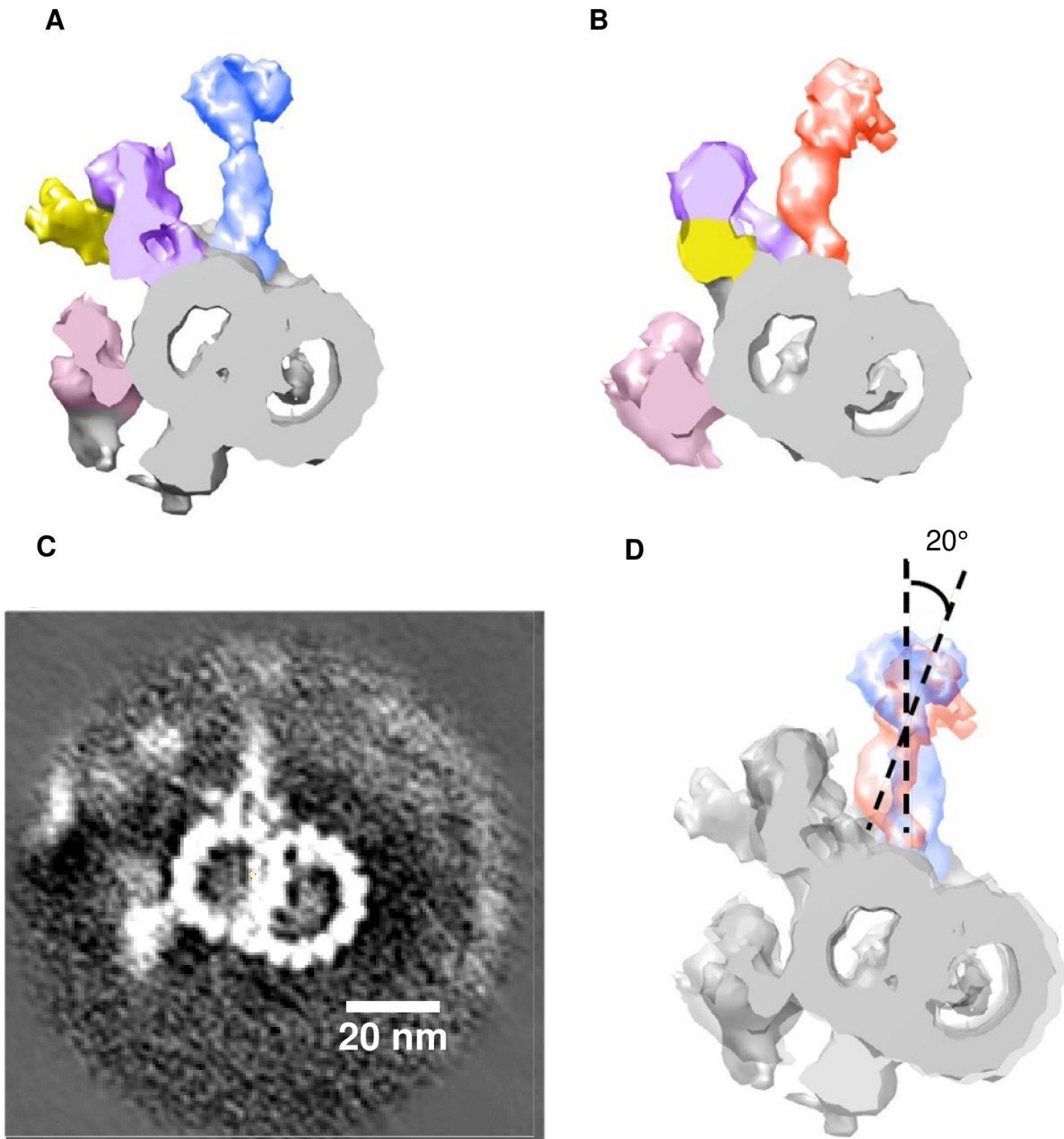

**Fig 3. Difference of angle of radial spoke stalk between RS2 and RS3. A-B**, Cross section of the doublet microtubules attached to the RS2 (A) and RS3 (B). **C**, Tomographic slice (5 nm-thick) of the doublet microtubule. **D**, RS2 superimposed with RS3. The stalk of RS3 shows a bending morphology. The RS2 head-stalk axis forms an angle of 20 degrees with the RS3 head-stalk axis.

*Chlamydomonas pf1* mutant with the *Rsp4* mutation, whereas the spoke head of RS3 remains in the human *RSPH1* mutation [17, 25]. Given that radial spoke head–deficient cells (*pf1*) are paralyzed in *Chlamydomonas* [13], a study has suggested that the remaining RS3 retains the motility of cilia in humans with *RSPH1* mutations [17]. Our data, however, suggest that the spoke heads of RS3 are not critical for the motility of the mouse cilia and that the axonemal dyneins and the doublet microtubules are sufficient for the generation of the circular motion of the mouse cilia. As a proof of this concept, eel sperm and mouse node cilia show rotational

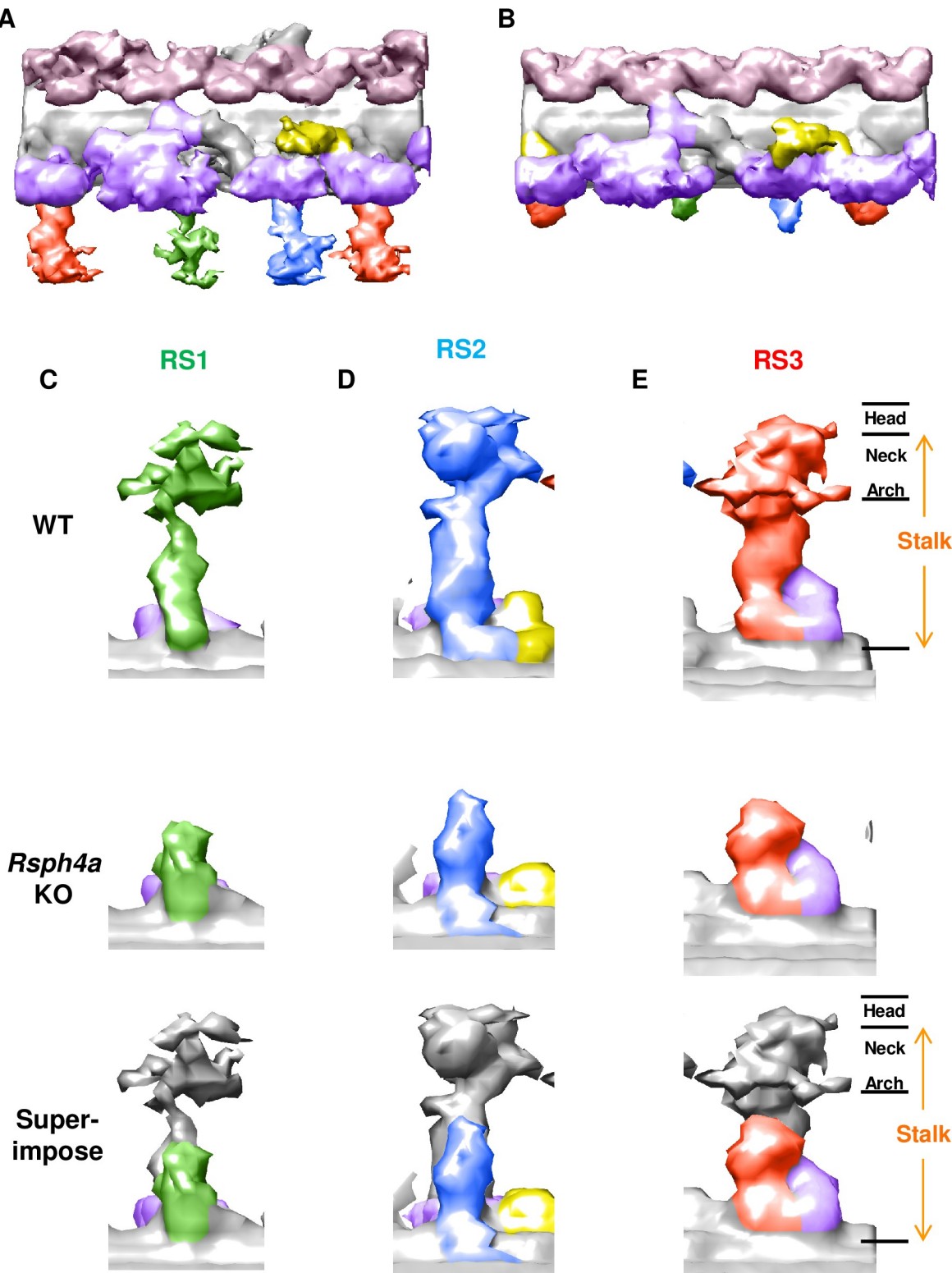

**Fig 4. Cryo-EM structure of the trachea cilia in Rsph4a KO mice. A-B**, The averaged structure of the axoneme repeat in WT mice (**A**) and Rsph4a KO mice (**B**). The triplet radial spoke heads are missing in the Rsph4a KO mice. **C**, Structure of each RSs. RSs consist of the spoke head and the stalk. The neck/arch is the most distal part of the stalk.

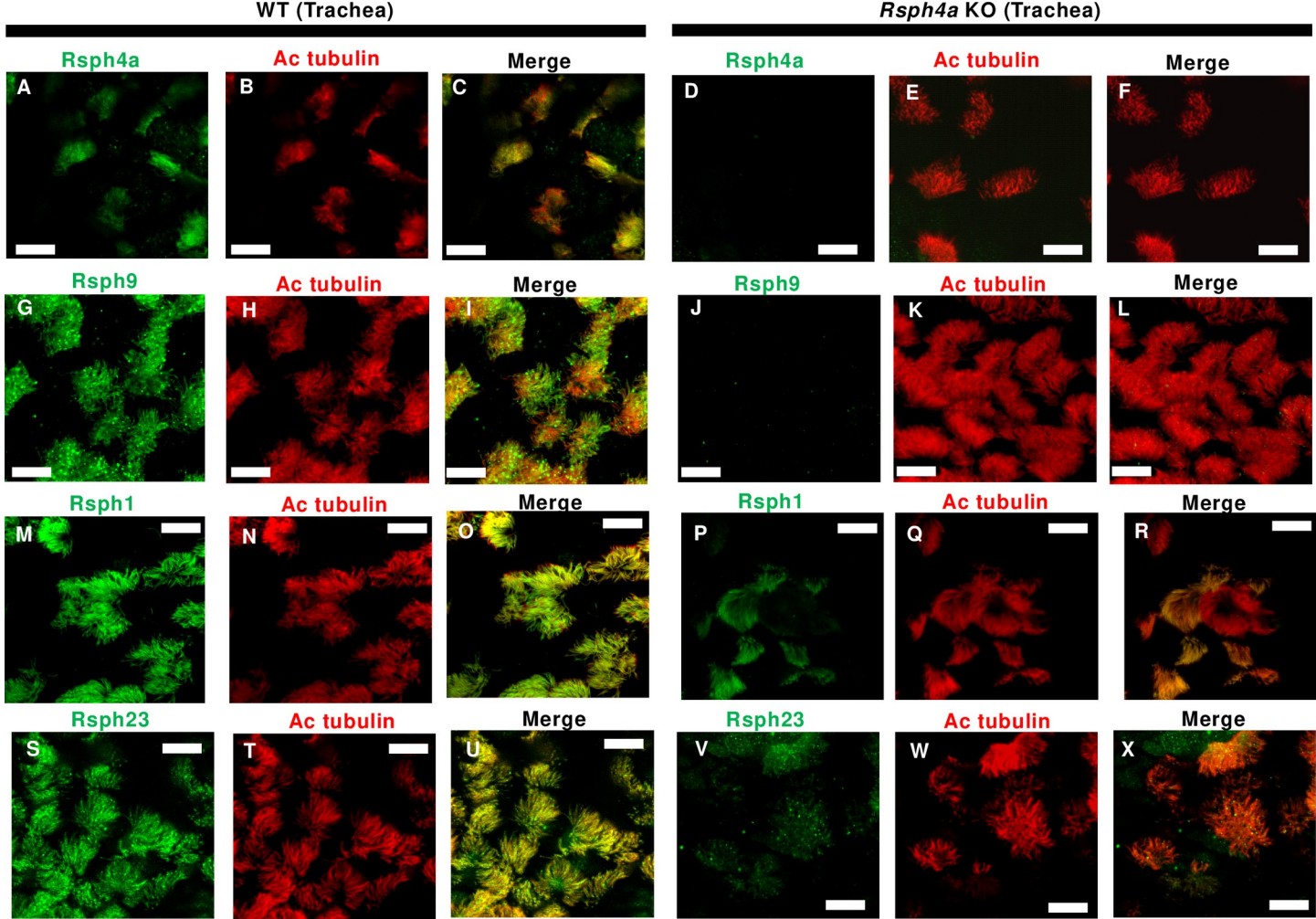

**Fig 5. Immunofluorescence analysis of radial spoke head proteins in the mouse trachea cilia. A-F,** Subcellular localization of Rsph4a and acetylated tubulin in the trachea cells in the WT (**A-C**) and Rsph4a KO (**D-F**) mice. Subcellular localization of Rsph9 and acetylated tubulin in the trachea cells in the WT (**G-I**) and Rsph4a KO (**J-L**) mice. Subcellular localization of Rsph1 and acetylated tubulin in the trachea cells in the WT (**M-O**) and Rsph4a KO (**P-R**) mice. Subcellular localization of Rsph23 and acetylated tubulin in the trachea cells in the WT (**S-U**) and Rsph4a KO (**V-X**) mice. All the size bars indicate 10 μm.

motion in the absence of RSs [29, 35, 36]. If so, what is the role of RS3? One possibility is that RS3 compensates for the function of RS1 and RS2. The proportion of respiratory cilia with normal axonemal structure is 50% in human *RSPH4A* patients [18], whereas it is 80% in *RSPH1* patients [16], suggesting that RSPH4A mutation causes more severe phenotype than RSPH1 mutation. RS3 alone may control the doublet microtubule arrangement inside the axoneme, and the functions of the triplet RSs seem to complement each other. Given that the spoke head of RS3 is retained in the human PCD patients with RSPH1 mutation [17], the lack of all the triplet spoke heads in Rsph4a KO mice could explain the more severe structural defect of axoneme of respiratory cilia in *RSPH4A* patients than *RSPH1* patients [16, 18].

Frommer et al. demonstrated that RSPH4A is the core radial spoke head protein of the human respiratory cilia by immunofluorescence [20]. Our data are consistent with this finding. We, in contrast, show that the spoke head and the neck/arch are also disrupted in the absence of Rsph4a in the mouse motile cilia. Rsp4/Rsph4a may act as a building block of the neck/arch [13], or the absence of a spoke head could destabilize the neck/arch complex in

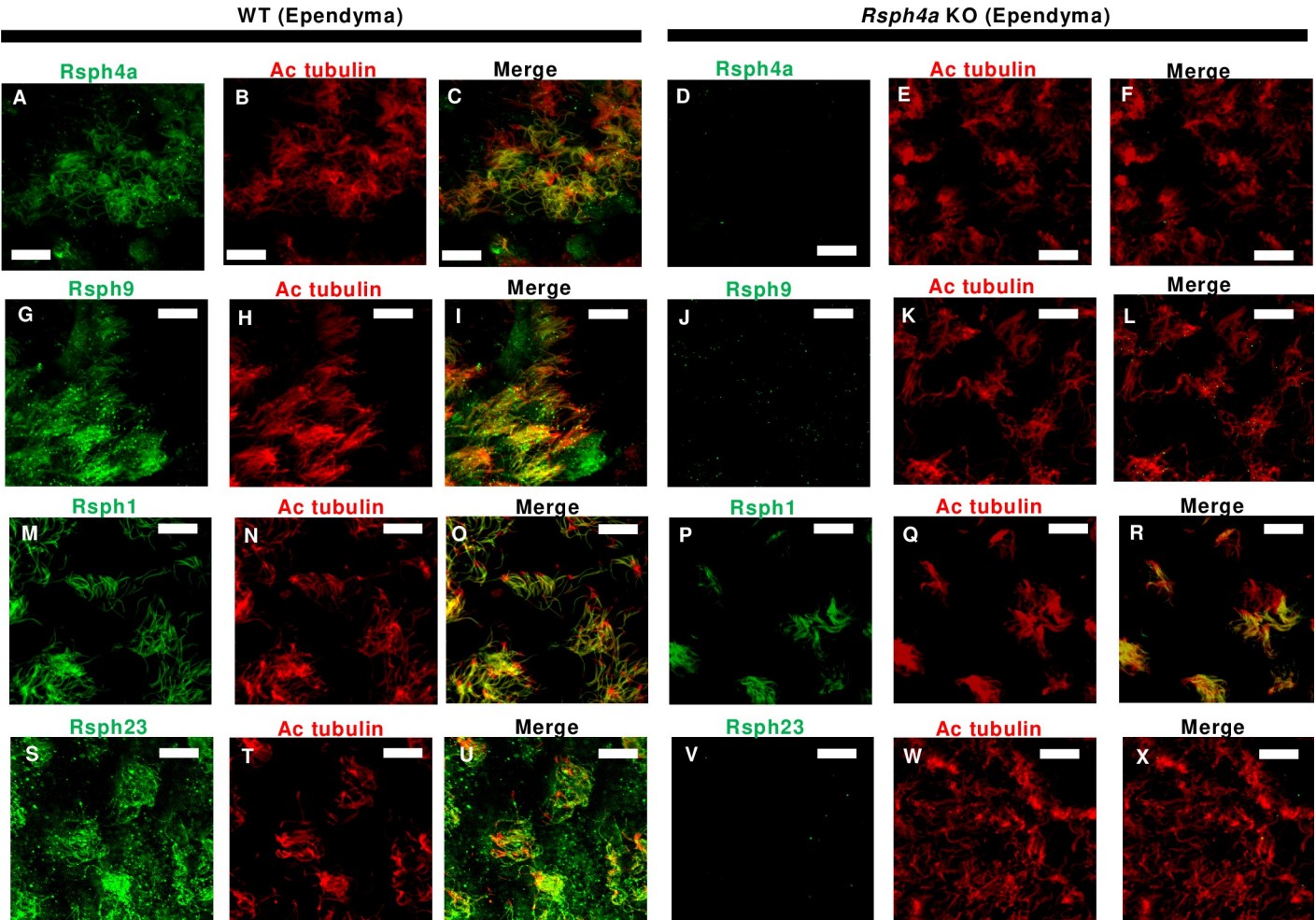

**Fig 6. Immunofluorescence analysis of radial spoke head proteins in the mouse ependymal cilia (Brain). A-F,** Subcellular localization of Rsph4a and acetylated tubulin in the ependymal cells in the WT (**A-C**) and Rsph4a KO (**D-F**) mice. Subcellular localization of Rsph9 and acetylated tubulin in the ependymal cells in the WT (**G-I**) and Rsph4a KO (**J-L**) mice. Subcellular localization of Rsph1 and acetylated tubulin in the ependymal cells in the WT (**M-O**) and Rsph4a KO (**P-R**) mice. Subcellular localization of Rsph23 and acetylated tubulin in the ependymal cells in the WT (**S-U**) and Rsph4a KO (**V-X**) mice. All the size bars indicate 10 μm.

mouse motile cilia. In previous works, Pigino et al. reported that the spoke heads were missing, whereas the neck/arch was retained in the *Chlamydomonas pf1* mutant (Rsp4 mutant) [25]. Frommer et al. reported that ciliary localization of the neck/arch protein RSPH23 was retained in respiratory tissue in human PCD patients with RSPH4A mutations [20]. The stability of the neck/arch may be different among species. Further investigation is necessary on the diversity of RSs and their physiological significance.

## Methods

### Animals

The mice were bred at the animal facility of the Bio-Resource Laboratory, Tokyo University of Agriculture & Technology, under a 12-h-light, 12-h-dark cycle and were provided with food and water ad libitum. All experiments were approved by the Institutional Animal Care and Use Committee of Tokyo University of Agriculture & Technology.

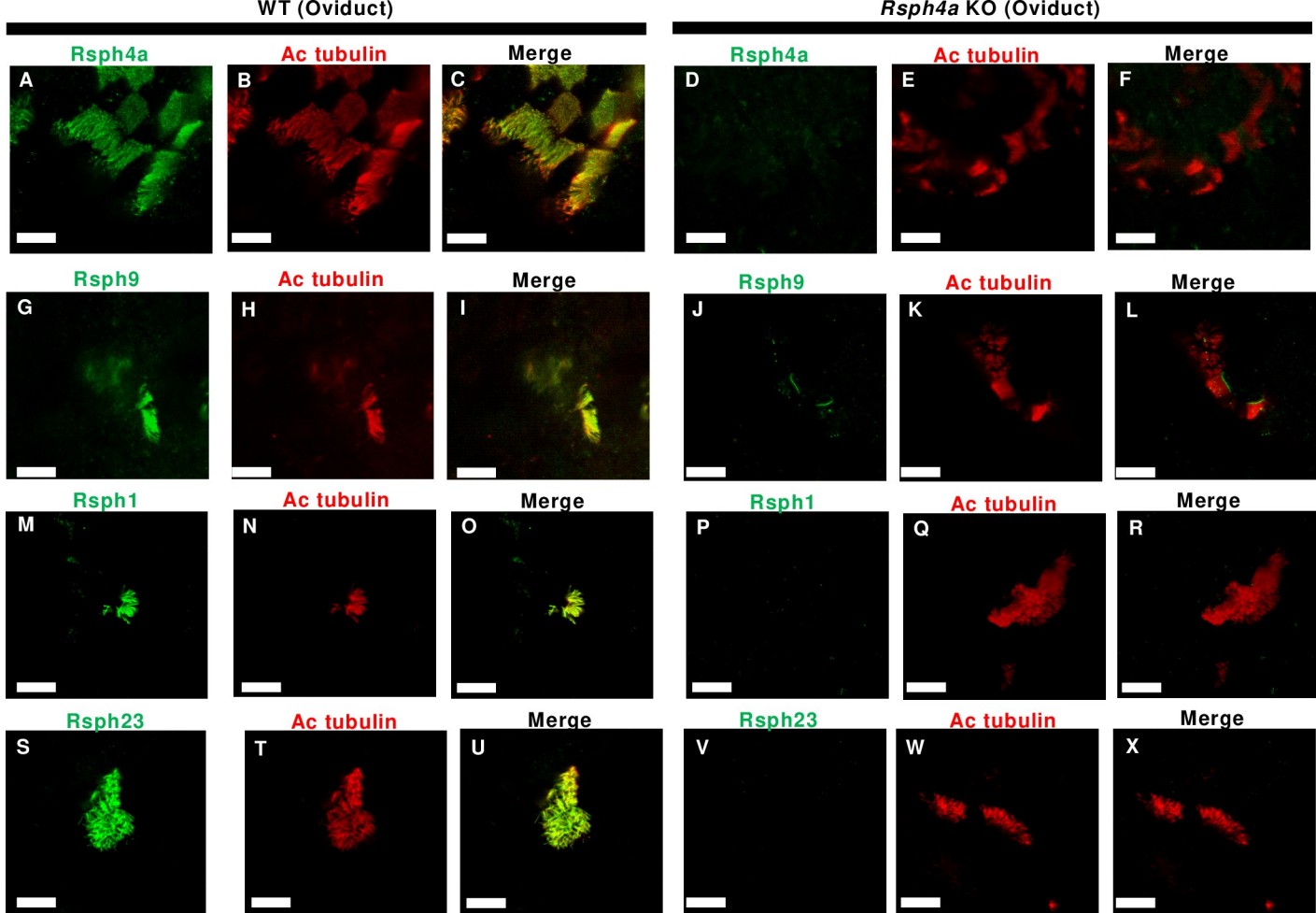

**Fig 7. Immunofluorescence analysis of radial spoke head proteins in the mouse oviduct cilia. A-F,** Subcellular localization of Rsph4a and acetylated tubulin in the oviduct cells in the WT (**A-C**) and Rsph4a KO (**D-F**) mice. Subcellular localization of Rsph9 and acetylated tubulin in the oviduct cells in the WT (**G-I**) and Rsph4a KO (**J-L**) mice. Subcellular localization of Rsph1 and acetylated tubulin in the oviduct cells in the WT (**M-O**) and Rsph4a KO (**P-R**) mice. Subcellular localization of Rsph23 and acetylated tubulin in the oviduct cells in the WT (**S-U**) and Rsph4a KO (**V-X**) mice. All the size bars indicate 10 μm.

## Generation of *Rsph4a*$^{-/-}$ mice

The design of the targeting vector is described in a previous work (S4 Fig in the paper from Ref. 30; Shinohara et al., 2015). *Rsph4a*$^{-/-}$ mice and control *Rsph4a*$^{+/+}$ (WT) littermates (C57B6J background) were generated by intercrossing *Rsph4a*$^{+/-}$heterozygotes. Polymerase chain reaction (PCR) primers for detection of the WT allele were 5´-CGAAAGCTTCGCAA TAAACA-3´ (P1) and 5´-CAGGGATACGAGGAACCAAA-3´ (P2), and those for detection of the *Rsph4a* knockout allele were 5´-CTCCATGGGCACTTACTTTC-3´ (P3) and P2.

## Immunofluorescence

The trachea, ependymal tissue, and oviduct tissue were dissected from mice on postnatal day 21 into phosphate-buffered saline, fixed for 10 minutes at room temperature with 4% parafor-maldehyde, and exposed to methanol at –20˚C for 3 minutes. The tissue was then incubated for 10 minutes at room temperature in a solution containing 0.1 M Tris-HCl (pH 7.5), 0.15 M NaCl, and 0.5% TSA blocking reagent (PerkinElmer) before incubation overnight at 4˚C with

rabbit antibodies to Rsph1 (HPA016816, Sigma, 1/100), Rsph4a (HPA031198, Sigma 1/100), Rsph9 (HPA031703, Sigma, 1/100), Rsph23 (HPA044555, Sigma, 1/100) and mouse antibodies to acetylated tubulin (T6793, Sigma, 1/200) diluted in blocking buffer. The samples were washed with phosphate-buffered saline containing 0.1% Triton X-100 and then incubated overnight at 4˚C with AlexaFluor-conjugated secondary antibodies (Life Technologies, 1/1000) diluted in blocking buffer. We used seven mice for Rsph4a assay, four mice for Rsph9, four mice for Rsph1, and five mice for Rsph23, respectively (We used the same number of wildtype and Rsph4a KO mice for each assay).

## Western blotting

The trachea and testis tissue were dissected from mice on postnatal day 56 into phosphate-buffered saline and we homogenized the tissue in urea/detergent mixture solution. We used Triton-X for the testis and NP40 for the trachea, respectively. After homogenization of tissues, we carried out centrifugation and collected supernatant as a lysate. For western blotting, we used the same antibody (dilution 1/1000) as well as the immunofluorescence. We used two wild type mice and three *Rsph4a* KO mice for the preparation of the trachea sample. In other hand, we used two wild type mice and two *Rsph4a* KO mice for the preparation of the testis sample.

## Imaging of ciliary motion

The trachea, ependymal tissue, and oviduct tissue were dissected from mice on postnatal day 21 into DMEM HEPES with 10% FBS. Three mice are used for the each observation (Three wild type mice and three *Rsph4a* KO mice). Tissue is set onto a slide glass with a silicon rubber spacer, and we put a 0.17 mm thick cover glass (Matsunami) on to the spacer before observation. The motion of cilia was captured for 5 s (200 frames/s for trachea cilia, 500 frames/s for ependymal cilia, and 200 frames/s for oviduct cilia) with a high-speed CMOS camera (HAS-500, Detect). The cells were observed by microscopy (Zeiss) equipped with a 100× oil-immersion objective lens for trachea/oviduct cilia and 60× water-immersion objective lens for ependymal cilia. The specimen was illuminated with transmitted light from a halogen lamp. Time-series images were captured at a resolution of 1024 by 992 pixels, with a pixel resolution of 0.082 by 0.082 μm.

## Cryoelectron tomography of mouse trachea cilia

For cryo-ET, the mouse trachea cilia samples were prepared according to a protocol in a previous work [30]. Four mice are used for the each observation (Four WT mice and four *Rsph4a* KO mice). Trachea is dissected from three weeks old mice (P21) in the PBS buffer. We placed the trachea tissue onto the wall of the 1.5 mL tube and rubbed it delicately in Tris buffer containing 5 mM DTT and then collected axonemes by centrifugation at 13,000 rpm for 15 minutes. Next, we carried out demembranation by treating the samples with 2% NP40 on ice for 1 hour followed by centrifugation at 13,000 rpm for 15 minutes. The samples were frozen in liquid ethane. Images were taken as described previously using a cryo-EM (Tecnai F20;FEI, Polara at Nagoya Univ.) equipped with a field emission gun, an energy filter, and a 4,092 × 4,092 charge-coupled device (Gatan). The accelerating voltage was set to 300 kV, and the magnification was set to 27,000 ×. Tomographic images in the range of ±55~70 degrees were acquired using Saxton scheme (~ 60 images in total) with 1 e$^-$ dose per Å$^2$ per one image, using Xplore3D software (FEI).

## Image processing (subtomogram averaging)

Tomogram reconstruction was performed using IMOD [37]. The subtomogram averaging procedures described below were performed using the electron microscope image analysis software program Eos [38], unless otherwise noted. First, low-resolution subtomograms with a pixel size of 50×50×36, which represent 96-nm structural repeat units from a doublet microtubule (with 36 pixels corresponding to 96 nm), were prepared from the tomograms that were shrunk to a quarter pixel size smaller (in each dimension) than the original ones. The low-resolution subtomograms were aligned and averaged using an averaged subtomogram from a sea urchin sperm axoneme as a reference for fitting. Then, high-resolution subtomograms representing a 96 nm repeat with a pixel size of 200×200×144 (with 144 pixels corresponding to 96 nm) were created from the original tomograms and were aligned and averaged using the averaged low-resolution subtomograms as a reference for fitting. Missing wedges were compensated in the averaging processes. A total of 322 particles from 4 tomograms were used for WT mouse cilia, and 491 particles from 4 tomograms were used for Rsph4a KO mouse cilia. For marking the positions of the axonemes or the doublet microtubules for cropping the images, a software program for image processing in structural biology, Bshow, in the Bsoft software package [39] was used. Tomographic slices were visualized with IMOD software (http://bio3d.colorado.edu/imod/index.html). Surface rendering, as well as denoising through hiding smaller blobs, binning and Gaussian filtering were performed with UCSF Chimera [40].

## Supporting information

**S1 Fig. Preparation of cryo-EM sample of mouse trachea cilia.** We dissected the mouse trachea and delicately rubbed it onto the wall of the tube to isolate cilia. Then, we collected the trachea cilia by ultracentrifugation, and the cilia were frozen in liquid ethane. Three-dimensional structures of the repeat unit of the axoneme are revealed by cryo-ET including cryoelectron microscope observation and subtomogram averaging (Methods).
(TIF)

**S2 Fig. Tomographic images of mouse trachea cilia.** We show the tomographic slice of the trachea cilia in the wild type mice (left) and in the *Rsph4a* KO mice (right). Bars are 20 nm.
(TIF)

**S3 Fig. Western blotting of spoke head and neck/arch protein.** We show western blotting data of spoke head protein (Rsph4a, Rsph1), and neck/arch protein (Rsph23). Rsph1 and Rsph23 are reduced in the trachea of *Rsph4a* KO mice.
(TIF)

**S1 Video. Motion of trachea cilia in the wildtype mouse.** The cilia show planar beating in the wildtype mouse. The speed is 10 frames/sec.
(AVI)

**S2 Video. Motion of trachea cilia in the *Rsph4a* KO mouse.** The cilia show clockwise rotation in the *Rsph4a* KO mouse. The speed is 10 frames/sec.
(AVI)

**S3 Video. Motion of ependymal cilia in the wildtype mouse.** The cilia show planar beating in the wildtype mouse. The speed is 20 frames/sec.
(AVI)

**S4 Video. Motion of ependymal cilia in the *Rsph4a* KO mouse.** The cilia show clockwise rotation mixed with planar beating in the *Rsph4a* KO mouse. The speed is 20 frames/sec.
(AVI)

**S5 Video. Motion of oviduct cilia in the wildtype mouse.** The cilia show planar beating in the wildtype mouse. The speed is 10 frames/sec.
(AVI)

**S6 Video. Motion of oviduct cilia in the *Rsph4a* KO mouse.** The cilia show anticlockwise rotation in the *Rsph4a* KO mouse. The speed is 10 frames/sec.
(AVI)

**S7 Video. Motion of oviduct cilia in the *Rsph4a* KO mouse.** The cilia show planar beating with small amplitude in the *Rsph4a* KO mouse. The speed is 10 frames/sec.
(AVI)

## Acknowledgments

We thank S. Dutcher for discussion on the structure of the radial spoke of mouse tracheal cilia. We also thank T. Ide, A. Fukumoto, H. Nishimura, Y. Ikawa, R. Negishi, and T. Yoshino for technical assistance with the generation of knockout mice and microscopy.

## Author Contributions

**Conceptualization:** Chikako Shingyoji, Kyosuke Shinohara.

**Data curation:** Hiroshi Yoke, Hironori Ueno, Akihiro Narita, Takafumi Sakai, Kahoru Horiuchi, Kyosuke Shinohara.

**Formal analysis:** Hiroshi Yoke, Hironori Ueno, Akihiro Narita, Kyosuke Shinohara.

**Funding acquisition:** Kyosuke Shinohara.

**Investigation:** Chikako Shingyoji.

**Methodology:** Hiroshi Yoke, Hironori Ueno, Akihiro Narita, Takafumi Sakai, Hiroshi Hamada, Kyosuke Shinohara.

**Project administration:** Kyosuke Shinohara.

**Resources:** Hiroshi Hamada, Kyosuke Shinohara.

**Software:** Hironori Ueno, Akihiro Narita.

**Supervision:** Kyosuke Shinohara.

**Validation:** Akihiro Narita, Kyosuke Shinohara.

**Visualization:** Hiroshi Yoke, Hironori Ueno, Akihiro Narita, Takafumi Sakai, Kahoru Horiuchi, Kyosuke Shinohara.

**Writing – original draft:** Hiroshi Yoke, Kyosuke Shinohara.

**Writing – review & editing:** Akihiro Narita, Kyosuke Shinohara.

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
