## [Decision Letter · Decision Letter 0]

3 Nov 2019

Dear Dr Yue-Qiu Tan,

Dear Dr.Qianjun Zhang

Thank you very much for submitting your Research Article entitled 'Rsph4a is essential for the triplet radial spoke head assembly of the mouse motile cilia' to PLOS Genetics. Your manuscript was fully evaluated at the editorial level and by independent peer reviewers. The reviewers appreciated the findings in regards to the molecular composition of radial spokes, but raised some substantial concerns about the current manuscript. Based on the reviews, we will not be able to accept this version of the manuscript, but we are happy to review again an improved version. We cannot, of course, promise publication at that time.

If you decide to revise the manuscript for further consideration at PLOS Genetics, please aim to resubmit within the next 60 days, unless it will take extra time to address the concerns of the reviewers, in which case we would appreciate an expected resubmission date by email to plosgenetics@plos.org.

[LINK]

We are sorry that we cannot be more positive about your manuscript at this stage. Please do not hesitate to contact us if you have any concerns or questions.

Yours sincerely,

Heymut Omran

Guest Editor

PLOS Genetics

Gregory Barsh

Editor-in-Chief

PLOS Genetics

Reviewer's Responses to Questions

**Comments to the Authors:**

Reviewer #1: This manuscript by Yoke et al. investigates the loss of radial spoke protein RSPH4A in mouse motile cilia structure. Although this gene was previously implicated in human primary ciliary dyskinesia and mouse ciliary motility, this study uses a combination of cryoelectron tomography and immunofluorescence to uncover the role of RSPH4A in radial spoke assembly. In addition, high-speed video microscopy also reveals subtle but distinct differences between ciliated cell types in the mouse. These data are novel and important, the methods are appropriate, and the conclusions are generally justified. However, the manuscript would benefit from addressing a few concerns prior to publication:

MAJOR COMMENTS:

1. FIGURE 1: Figure 1A is not discussed in the manuscript. What is the gross phenotype of these mice?

2. FIGURE 2: Figure 2D is not discussed in the manuscript.

3. FIGURES 5, 6, and 7: The authors state that RSPH9 and RSPH1 are “dramatically reduced” in all mutant ciliated cell types. However, while RSPH9 appears to be reduced, RSPH1 looks very similar to wild type in the tracheal epithelial and ependymal cells (Fig. 5P, 6P). Similarly, the authors state that RSPH23 is not present in cilia from any of the cell types, but it appears to be present in the trachea (Fig. 5V). How consistently are these proteins reduced in the mutant cells? Could fields that are more representative be shown in these figures? Quantitative western blotting could also potentially address the difference in protein levels that might be difficult to see by immunofluorescence.

4. INTRODUCTION, 1st paragraph: The authors describe the ciliary motility defect in patients with mutations in RSPH1, RSPH4, and RSPH9, but it would be helpful if they also discussed the ultrastructural defects in the cilia from these patients, as that likely contributes to the abnormal beating patterns. In addition, human PCD patients have been reported with mutations in RSPH3 (Jeanson et al., 2015), which should be referenced.

5. METHODS: Some important information is missing from the Methods Section. The following should be added:

• Generation of Rsph4a-/- mice: The genetic background of the mutant mice

• Immunofluorescence: The number of mice used for each experiment, the concentration or dilution factor for each primary antibody, the specific secondary antibodies

• Imaging of ciliary motion: The age and number of mice used for the experiment, the method of tissue collection and preparation

• Cryoelectron tomography: The age and number of mice used for the experiment

MINOR COMMENTS:

1. INTRODUCTION, 1st paragraph: The authors state that “multiple motile cilia exist in the trachea, brain/ependymal, oviduct, inner ear, nasal, testis, and so on.” The phrase “and so on” should be specified, as some readers may not know the exact locations of motile cilia in the body.

2. DISCUSSION, 2nd paragraph: The authors state that “the stability of the neck/arch is most likely different among species.” This is an important point, but is it possible that assembly of the neck/arch could be different as well?

3. METHODS, Generation of Rsph4a-/- mice: The design of the targeting vector needs an appropriately formatted reference.

Reviewer #2: Review

'Rsph4a is essential for the triplet radial spoke head assembly of the mouse motile cilia' (PGENETICS-D-19-01560)

Recent cryoelectron tomography data reveal three types of radial spokes (RS1, RS2, and RS3) in the 96 nm axoneme repeat unit; however, the molecular composition of the third radial spoke, RS3 is unknown. Here, the authors describe that Primary Ciliary Dyskinesia protein Rsph4a plays critical role in the radial spoke head assembly not only of RS1 and RS2 but also of RS3. Examination of the cryoelectron tomography structure and the immunofluorescence analyses of wild type and Rsph4a-deficient mutant mice led them to conclude that Rsph4a is a generic spoke head protein of the triplet radial spoke in mouse motile cilia.

The manuscript is nicely written and well understandable.

The main and novel claim, that Rsph4a plays critical role in the radial spoke head assembly not only of RS1 and RS2 but also of RS3 is significant both for the field of basic cilia biology and the diagnostic field, because it provides additional information about cilia structure and an explanation of the diverse phenotype of patients with Primary Ciliary Dyskinesia (PCD) caused by radial spoke defects.

This finding is nicely demonstrated and supported by cryo-EM data of trachea of WT and Rsph4 mutant mice. However, there are major points to consider prior to publication and additional information is needed prior to publication:

1. Page 5: “however, the ciliary localization of Rsph9 and Rsph1 was dramatically reduced in the tissues (Fig. 5 J-L and P-R, Fig. 6 J-L 6 and P-R, Fig. 7 J-L and P-R)”

Please provide figures with higher quality. Especially acetylated tubulin appears to be overexposed.

a) Please quantify reduction compared to control; how do you explain the reduction, taking into account your cryo-EM results of trachea that suggest complete loss of at least the head, neck and arch?

b) Do you see differences in cryo-EM results between tracheal cilia, ependymal cilia, oviduct? Differences in your IF result suggest so. If not already done, cryo-EM on those additional cilia would improve your data and conclusions.

c) How do you explain differences in localization of especially Rsph1 and Rsph23 in trachea, ependymal cells, oviduct? Especially the normal appearing Rsph1 localization in trachea and ependyma of Rsph4 mutant mice?

d) Rsph1 staining in Figure 7 appears to be basal body staining, please comment

2. Page 6: “Conversely, Rsph23 was not localized in the axoneme of the motile cilia in the trachea, ependymal tissues, and the oviduct tissues in Rsph4a KO mice (Fig. 5 V-X, Fig. 6 V-X, Fig. 7 V-X)”: Rsph23 is detected in cilia of Rsph4 KO trachea as shown in Figure 5. Please correct and comment.

3. Materials and Methods, Imaging of ciliary motion: Information about sperm has not been provided, but mentioned in M&M. Please correct or include data.

4. Supplementary videos 1,5,6 appear to be different format and cannot be assessed.

5. Please use proper nomenclature for proteins in mouse

Reviewer #3: The functional relationships between the structural organization of the cilia and their beating behavior is a fascinating biological question. Recent advances in cryo-electron tomography observations allowed major breakthroughs in understanding the architecture of the different components involved in ciliary motility.

In this manuscript, the authors analyze the role of the radial spoke protein Rsph4 in mouse multiple ciliated epithelia. In all three type of Rsph4-/- multiple ciliated epithelia, ciliary motility is affected, with striking differences between tissues. For the first time the authors describe the ultrastructural organization of the radial spokes in mouse ciliated airways. They show that like in other vertebrates, 3 different radial spokes can be observed in the 96 nm axonemal repeats. Interestingly, they observed marked differences with humans in the architecture of the spoke head and in the connections between Radial spokes 2 and 3. They also show that Rsph4a is, a key component required for head formation of the three types of radial spokes. Last the authors show that Rsph4 is, like in humans, required for the assembly of Rsph1, Rsph9 at radial spokes but also, unlike in humans, of Rsph23 required for neck/arch assembly.

This manuscript thus highlights evolutionary divergences between species and pave the way to understand the origin of the diversity of cilia motile properties.

I only have a few minor concerns regarding the presentation of the data:

-Could the authors present at least one or two examples of the averaged tomographic images and not just the surface rendering of it?

-I am not sure to understand the description of the differences between human and mouse. Human radial spoke heads were compared to pairs of ice blade in Lin et al 2014, and on panel 2B mouse radial spoke heads do not show such parallel “blades”? The text and conclusions are confusing compared to the images and with the initial description of human radial spokes in Lin et al.

-The movie of control oviduct cilia is hard to compare to the rsph4 mutant ones, maybe another movie could help to better see the differences in motility?

**Have all data underlying the figures and results presented in the manuscript been provided?**

Reviewer #1: Yes

Reviewer #2: Yes

Reviewer #3: Yes

PLOS authors have the option to publish the peer review history of their article (what does this mean?). If published, this will include your full peer review and any attached files.

Reviewer #1: No

Reviewer #2: No

Reviewer #3: No

---

## [Editor Report · Decision Letter 1]

12 Feb 2020

Dear Dr Shinohara,

We are pleased to inform you that your manuscript entitled "Rsph4a is essential for the triplet radial spoke head assembly of the mouse motile cilia" has been editorially accepted for publication in PLOS Genetics. Congratulations!

Yours sincerely,

Heymut Omran

Guest Editor

PLOS Genetics

Gregory Barsh

Editor-in-Chief

PLOS Genetics

Comments from the reviewers (if applicable):

**Data Deposition**

http://datadryad.org/submit?journalID=pgenetics&manu=PGENETICS-D-19-01560R1

**Press Queries**

---

## [Editor Report · Acceptance letter]

16 Mar 2020

PGENETICS-D-19-01560R1 

Rsph4a is essential for the triplet radial spoke head assembly of the mouse motile cilia 

Dear Dr Shinohara, 

We are pleased to inform you that your manuscript entitled "Rsph4a is essential for the triplet radial spoke head assembly of the mouse motile cilia" has been formally accepted for publication in PLOS Genetics! Your manuscript is now with our production department and you will be notified of the publication date in due course.

With kind regards,

Matt Lyles

PLOS Genetics

On behalf of:
